# Thermal and electrical signatures of a hydrodynamic electron fluid in tungsten diphosphide

J. Gooth[1,2], F. Menges[1,4], N. Kumar[2], V. Süβ[2], C. Shekhar [iD][2], Y. Sun [iD][2], U. Drechsler[1], R. Zierold[3], C. Felser [iD][2] & B. Gotsmann[1]

In stark contrast to ordinary metals, in materials in which electrons strongly interact with each other or with phonons, electron transport is thought to resemble the flow of viscous fluids. Despite their differences, it is predicted that transport in both conventional and correlated materials is fundamentally limited by the uncertainty principle applied to energy dissipation. Here we report the observation of experimental signatures of hydrodynamic electron flow in the Weyl semimetal tungsten diphosphide. Using thermal and magneto-electric transport experiments, we find indications of the transition from a conventional metallic state at higher temperatures to a hydrodynamic electron fluid below 20 K. The hydrodynamic regime is characterized by a viscosity-induced dependence of the electrical resistivity on the sample width and by a strong violation of the Wiedemann–Franz law. Following the uncertainty principle, both electrical and thermal transport are bound by the quantum indeterminacy, independent of the underlying transport regime.

[1] IBM Research—Zurich, Säumerstrasse 4, 8803 Rüschlikon, Switzerland. [2] Max Planck Institute for Chemical Physics of Solids, Nöthnitzer Straße 40, 01187 Dresden, Germany. [3] Institute of Nanostructure and Solid-State Physics, Universität Hamburg, Jungiusstraße 11, 20355 Hamburg, Germany. [4] Present address: Department of Physics, University of Colorado Boulder, Boulder 80309-0390 CO, USA. Correspondence and requests for materials should be addressed to J.G. (email: johannes.gooth@cpfs.mpg.de) or to B.G. (email: bgo@zurich.ibm.com)

In an overwhelmingly large group of conducting materials, charge transport can be described by the rather simple model of a free-electron gas. Its basis is that the carriers move unimpededly until they scatter with phonons or defects. Such collisions usually relax both the momentum and the energy currents, and consequently impose a resistance to charge and heat flow alike. In most conventional conductors, electrical and thermal transport are therefore related via the Wiedemann–Franz law, which states that the product of the electrical resistivity $\rho$ and the thermal conductivity $\kappa$, divided by the temperature $T$ is a constant $L = \rho\kappa/T$, yielding the Sommerfeld value $L_0 = 2.44 \times 10^{-8}$ W $\Omega$ K$^{-2}$. $\rho$ and $\kappa$ are intrinsic material properties and independent of the size and geometry of the conducting bulk. However, the conventional free-electron model fails to describe transport in strongly interacting electron systems[1]. The difficulty is to find a theoretical framework that captures the frequent inter-particle collisions that define the interaction within the many-body system. Recently, it has been rediscovered that the theory of hydrodynamics, which is normally applied to explain the behavior of classical liquids like water, could be used to describe the collective motion of electrons in such a system[2–9].

In contrast to the free-electron gas, the energy dissipation in a hydrodynamic electron fluid is dominated by momentum-conserving electron–electron scattering or small-angle electron–phonon scattering[10]. A signature of the hydrodynamic nature of transport emerges when the flow of the electrons is restricted to channels[11,12]. The electrical resistance of a hydrodynamic electron liquid is then proportional to its shear viscosity, and therefore paradoxically increases with increasing mean free path of the electrons[2,13]. Viscosity-induced shear forces at the channel walls cause a nonuniform velocity profile, so that the electrical resistivity becomes a function of the channel width. Moreover, the electrical resistivity will become small with increasing width, because momentum-relaxing processes within the bulk are strongly suppressed. The thermal conductivity is instead dominated by faster momentum-conserving collisions. Consequently, a strong violation of the Wiedemann–Franz law is predicted[3–5,14].

Despite the significant difference in the microscopic mechanisms behind momentum- and energy-current-relaxing collisions, both processes should be limited by the quantum indeterminacy in the energy dissipation with a timescale larger than $\tau_\hbar = \hbar/(k_B T)$[3,15–17], where $\tau_\hbar$ is determined only by the Boltzmann constant $k_B$, the reduced Planck constant $\hbar$ and the temperature $T$. This concept of, sometimes called Planckian dissipation, follows directly from the uncertainty principle when one applies equipartition of energy and any degree of freedom only carries $k_B T$[3].

For quantum hydrodynamic fluids, it has been developed in the frameworks of a string theory, known as anti-de Sitter space/conformal field theory correspondence (AdS/CFT)[1,18]. Holographic models successfully predicted the universal bound on the momentum-relaxation time $\tau_{mr}$ in a strongly interacting neutral plasma[19]. The momentum-relaxation bound $\tau_{mr} \geq \tau_\hbar$ can also be expressed as the ratio of the shear viscosity to the entropy density, and is not only supported by experiments on the quark-gluon plasma and on the ultracold Fermi gas[20,21], but is also well-respected in classical fluids such as water.

Likewise, maximally dissipative processes matching the time-scale $\tau_\hbar$ have recently been proposed to underpin the $T$-proportional resistivity of metals[3,15], and are believed to be at the root of high-temperature superconductivity[17,22]. However, whether the AdS/CFT predictions prove to be relevant and provide a true bound for hydrodynamic electron systems is still an open question. In the context of charge transport described by quasi-particles, $\tau_{mr}$ represents the characteristic scattering time to

randomize the excess forward momentum of a quasi-particle. Momentum relaxation in most conductors is determined by the way in which the electrons couple via Umklapp processes to the lattice or to the disorder of the host solid. Thus, $\tau_{mr}$ of the electron system is determined by extrinsic coupling parameters and is not generally universal[3]. In the hydrodynamic regime, the momentum and energy-current relaxations are independent processes, which in principle enables the isolation of the momentum-conserving scattering time $\tau_{mc}$. The characteristic time $\tau_{er}$ needed to dissipate the excess energy of a quasi-particle includes both momentum-relaxing and conserving scattering processes. As such, to a first approximation the momentum-conserving scattering time is $1/\tau_{er} = 1/\tau_{mc} + 1/\tau_{mr}$[5].

Hydrodynamic effects have been postulated to have a role in the $T$-linear resistivity of high-temperature superconductors above their critical temperature, but purely hydrodynamic transport is not directly applicable to most of those systems. Extracting $\tau_{mr}$ and $\tau_{er}$ separately from experimental data has remained challenging because of the strong momentum-relaxation contribution. In fact, as momentum-relaxation processes are always present in a real material system, momentum can only be quasi-conserved. This, however, does not mean that hydrodynamic signatures are not observable in transport experiments. Hydrodynamic effects become dominant, when the momentum-conserving scattering length $l_{mc} = v_F \tau_{mc}$ provides the smallest spatial scale in the system, $l_{mc} \ll w \ll l_{mr}$, where $w$ is the sample width, $l_{mr} = v_F \tau_{mr}$ the momentum-relaxing scattering length and $v_F$ the Fermi velocity[11,23].

Inspired by pioneering experiments on semiconductor wires[24], signatures for hydrodynamic electron flow were recently reported in ultraclean PdCoO$_2$[23] and graphene[25–27]. However, as these materials exhibit only relatively weak scattering, experimental evidence of a universal thermal dissipation bound has been elusive. It is therefore desirable to go beyond previous experiments and investigate the dissipative timescales of a material in the limit of strong hydrodynamics.

Here we report the observation of experimental signatures of hydrodynamic electron flow in the Weyl semimetal tungsten diphosphide (WP$_2$). Using thermal and magneto-electric transport experiments, we find indications of the transition from a conventional metallic state at higher temperatures to a hydrodynamic electron fluid below 20 K. The hydrodynamic regime is characterized by a viscosity-induced dependence of the electrical resistivity on the sample width and by a strong violation of the Wiedemann–Franz law. Following the uncertainty principle, both electrical and thermal transport are bound by the quantum indeterminacy, independent of the underlying transport regime.

## Results

**Tungsten diphosphide bulk characterization**. For our study, we have chosen the semimetal WP$_2$[28] because momentum-relaxing Umklapp scattering is anomalously strongly suppressed in this material. WP$_2$ exhibits a space-group symmetry with two pairs of twofold degenerated Weyl points close to the intrinsic Fermi level. WP$_2$ contains a mirror plane perpendicular to the $a$-axis, a $c$-glide perpendicular to the $b$-axis and a twofold screw axis along the $c$-axis. Owing to the high crystalline anisotropy, typical WP$_2$ crystals are needle-shaped with an orientation along the $a$-axis. Moreover, the magneto-transport in bulk single crystals has been shown to be highly anisotropic by 2.5 orders of magnitudes between the $a-c$ and the $a-b$ plane[28], similarly to bulk PdCoO$_2$[29]. We note that although WP$_2$ is referred to as a semi-metal, it exhibits a finite density of free charge carriers at the Fermi energy (Supplementary Fig. 1). The best single-crystalline bulk samples of WP$_2$ exhibit a $T$-linear electrical resistivity above

150 K that is dominated by electron–phonon Umklapp scattering and a temperature-independent resistivity below 20 K as previously observed in $PdCoO_2$[23]. Its residual resistivity is only 3 nΩ cm, i.e., a remarkable three times lower value than that of $PdCoO_2$ in the hydrodynamic regime. At temperatures between 20 and 150 K, $\rho$ increases exponentially with increasing $T$, which has been attributed to the phonon-drag effect. Phonon drag is considered beneficial for reaching the hydrodynamic regime, because it provides another source of momentum-conserving scattering[23]. At 4 K, the bulk samples exhibit a mean free path of $l_{mr} \approx 100$ μm in the $a$–$c$ direction (see Supplementary Note 1 and Supplementary Figure 2 for details). The $l_{mr}$ of $WP_2$ exceeds the momentum-relaxing scattering length of hydrodynamic $PdCoO_2$ and that of graphene[25,26] by one and two orders of magnitude, respectively. These properties make $WP_2$ an ideal material for investigating hydrodynamic effects and the associated dissipative bounds in its strongly correlated electron system.

**Microbeam characterization**. For our experiments, we produced a series of $WP_2$ micro-ribbons by milling chemical vapor transport grown single crystals. The high anisotropy of the crystals results in rectangular micro-samples that retain the needle-shaped orientation of the grown crystal. Using a micro-manipulator, the milled micro-ribbons were transferred to a pre-defined metallic line structure (Fig. 1a). The solely mechanical fabrication method prevented chemical contamination and damage of the ultra-pure source material. Electron-beam deposition of platinum was used to provide electrical contacts by connecting the ribbon

ends to the underlying metal lines. Electron microscopy was used to determine the distance between the contact lines, i.e., the length $l$ of the micro-ribbon along the $a$-axis of the crystal, its thickness $t$ along the $b$-axis, and its width $w$ along the $c$-axis. The transport direction in our samples matches the crystal's $a$-axis. The thickness is approximately $t = w/2$. However, the high anisotropy of the magneto-transport yields a mean free path in the $a$–$b$ direction that is about 250 times lower than that in the $a$–$c$ direction[29]. Therefore, $w$ is the characteristic length scale of the samples[28], justifying the use of two-dimensional models for the in-plane transport properties in $a$–$c$. We investigated four micro-ribbons with widths of 0.4, 2.5, 5.6, and 9 μm, all satisfying $w \ll l_{mr}$ at low temperatures. Note that to observe the surface effects of all walls in equal strength in the transport experiments, one would have to design the sample such that its width is 250 times larger than its height.

**Width-dependence of the electrical resistivity**. In a first set of transport experiments, we studied the $T$- and $w$-dependence of the electrical resistivity $\rho = V/I \cdot wt/l$ of the micro-ribbons (Fig. 1b). For this purpose, we measured the voltage response $V$ to an AC-current bias $I$, using the standard low-frequency lock-in technique (see Methods for details). The elongated geometry of the micro-ribbons with contact lines wrapping around the whole cross section of the samples was chosen to ensure homogenous current distributions. Because of the low resistivity of the bulk sample, special care must be taken in extracting the intrinsic $\rho$ of the $WP_2$ ribbons. We therefore compare four-terminal with

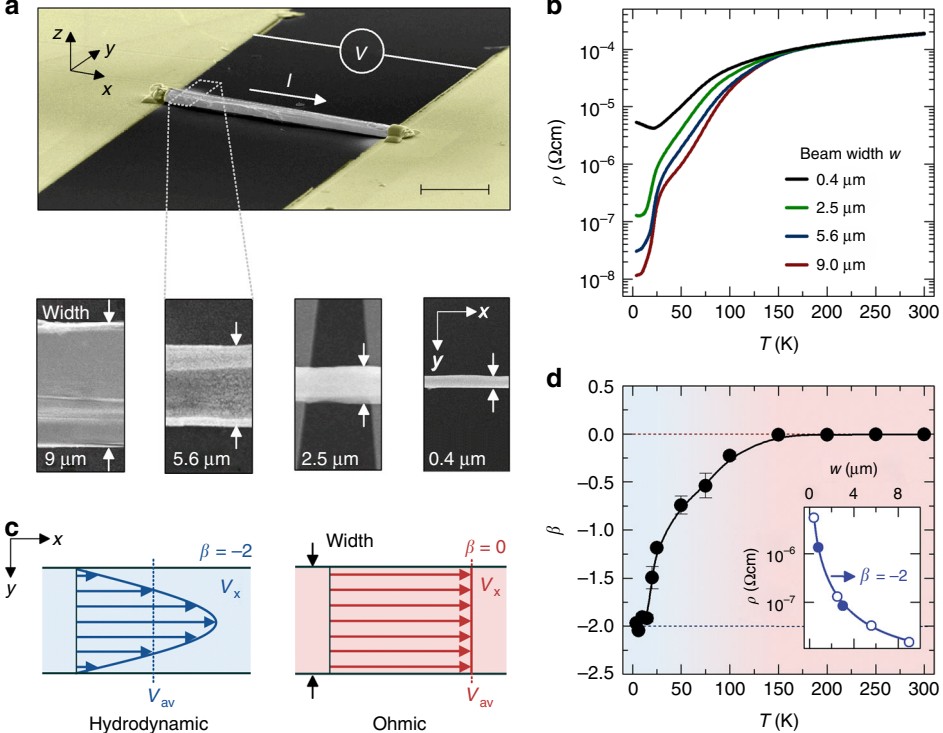

**Fig. 1** Effect of the channel width on the electrical resistivity. **a** False-colored scanning electron-beam microscopy (SEM) image of the device to measure the electrical resistance $R = V/I$ (upper panel) of the $WP_2$ micro-ribbons, where $I$ is the applied current and $V$ the measured voltage. $x$, $y$, and $z$ are the spatial dimensions. The scale bar denotes 10 μm. Ribbons of four different widths $w$ were investigated (lower panels). The error of the measured width is below 5%, including the uncertainty of the SEM and sample roughness. **b** Temperature ($T$)-dependent electrical resistivity $\rho$ of the four ribbons. **c** Sketch of the velocity $v_x$ flow profile for a Stokes flow (left panel, blue line and narrows) and conventional charge flow (right panel, red line and arrows). $v_{av}$ is the average velocity of the charge-carrier system. $\beta$ is the exponent of the functional dependence $\rho(w) = \rho_0 + \rho_1 w^\beta$. **d** $\beta$ as a function of $T$, extracted from power-law fits of the data plotted in **b**. The error bars denote the errors of the fits. The inset shows a power-law fit at 4 K, where the open and filled symbols represent quasi-four and four-terminal measurements, respectively. The line is a power-law fit, leading to $\beta = -2$. The colored background marks the hydrodynamic (light blue) and the normal metallic (Ohmic) temperature regime (light red)

quasi-four-terminal resistivity measurements. The quasi-four-terminal measurements exclude the resistance of the contact lines, but in principle include the interface resistances at the metal/semimetal junction. The electrical contact resistances are found, however, to be negligibly small, and not measurable in our experiment (see Fig. 1d and the Supplementary Note 2 and Supplementary Figure 3–6 for a detailed analysis). Conventionally, $\rho = \rho_0 = m^\star/(e^2 n \tau_{mr})$ is an intrinsic bulk property and does not depend on $w$. According to the Drude model, $\rho$ only depends on the effective mass $m^\star$ of the charge carriers, the elementary charge $e$ and the carrier concentration $n$. However, when boundary scattering becomes significant, $\rho$ can turn into a power-law function of the sample size $\rho \sim w^{\beta}$[27]. The exponent $\beta$ characterizes different transport regimes. In the well-established ballistic regime ($w \ll l_{er}, l_{mr}$), for example, a power of $0 > \beta \geq -1$ occurs and the electrical resistivity in the limit of fully diffusive scattering is given by is given by $\rho \sim w^{-1}$. Further, in a hydrodynamic fluid ($l_{er} \ll w \ll l_{mr}$), the flow resistance is determined solely by the interaction with the sample boundaries, reducing the average flow velocity of the electron fluid (Fig. 1c). As a consequence, a power of $-1 > \beta \geq -2$ is indicative of hydrodynamics. Recent theory predicts that the electrical resistivity in the Navier–Stokes flow limit is modified as $\rho = m^\star/(e^2 n) \cdot 12 \eta w^{-2}$, where electron–electron and small-angle electron–phonon scattering are parameterized in the shear viscosity $\eta$ (6−8).

As shown in Fig. 1b, all ribbons investigated consistently exhibit a constant $\rho = \rho_0$ above 150 K. In accordance with the bulk measurements[28], $\rho$ increases linearly with increasing $T$, as expected for dominant electron–phonon scattering. At lower temperatures, however, $\rho$ becomes a non-monotonic function of $T$ and increases with decreasing $w$. The change of slope in $\rho(T)$ is more pronounced in narrower ribbons, corroborating the importance of the sample's spatial boundaries in this temperature regime. As in real materials, the momentum is only quasi-conserved, $\rho$ always contains a width-independent Drude offset $\rho_0$ in addition to the width-dependent power-law component $\rho_1 w^{\beta}$. To extract the power $\beta$ from the experimental data (Fig. 1d and Supplementary Fig. 4), we have subtracted $\rho_0$ from $\rho$ at all temperatures, fitting the experimental data with $\rho = \rho_0 + \rho_1 w^{\beta}$. The exponents obtained were then cross-checked by a logarithmic analysis of $\rho - \rho_0 = \rho_1 w^{\beta}$ (Supplementary Fig. 7 and Supplementary Note 3). As shown in Fig. 1d, we found that the low-temperature regime ($T < 20$ K) is well described by an inverse quadratic relation $\rho - \rho_0 \sim w^{-2}$, in agreement with the Navier–Stokes description of hydrodynamic flow[11–13]. The residual resistivity $\rho_0$ of about 4 nΩ cm obtained at 4 K matches the bulk resistivity excellently.

This result enables a quantitative extraction of the kinematic shear viscosity of the electron liquid as $\eta = 3.8 \times 10^{-2}$ m² s⁻¹ at 4 K (see Supplementary Note 3). Multiplication by the mass density $M = nm^\star$ yields a dynamic viscosity of about $\eta_D = 1 \times 10^{-4}$ kg m⁻¹ s⁻¹ at 4 K, which is on the order of that of liquid nitrogen at 75 K. The observed $w^{-2}$ dependence provides strong evidence of hydrodynamic electron flow in WP₂, whereas the regime between 20 and 150 K can be explained by a smooth transition to a hybrid state in which viscosity-stimulated boundary scattering mixes with momentum-relaxing electron–phonon collisions[23].

## Estimation of the Lorenz number

Next, we investigated the Lorenz number $L = \kappa \rho / T$ in WP₂, which is widely considered to be an important observable for characterizing thermal and charge transport characteristics[5,14,26,30]. Therefore, we determined the thermal conductivity $\kappa$ of the 2.5-μm-wide WP₂ sample. The measurements were performed with open boundary conditions, prohibiting electric current flow (see Methods for details). Zero

electric current forces zero momentum flux, which decouples the heat flow from the momentum drag in the hydrodynamic regime[14]. The ribbon was mounted on a microsystem platform[30,31] with two integrated heater/sensors as shown in Fig. 2a. The sensor device is thermally insulated through 1.2-mm-long silicon nitride bars operated in vacuum to $1.6 \times 10^{-5}$ K W⁻¹ at room temperature. The fabrication and characterization of the platform are described in detail in the Methods and in the Supplementary Figures 8–10. Two gold resistors, serving as both thermometers and heaters, were calibrated at each temperature and used to measure both the temperature bias along and the heat flux through the WP₂ sample. Although used routinely for the thermal characterization of microscale and nanoscale samples, this method often suffers from thermal contact resistance effects, in particular when applied to the characterization of nanoscale structures. To minimize such effects, we chose large dimensions

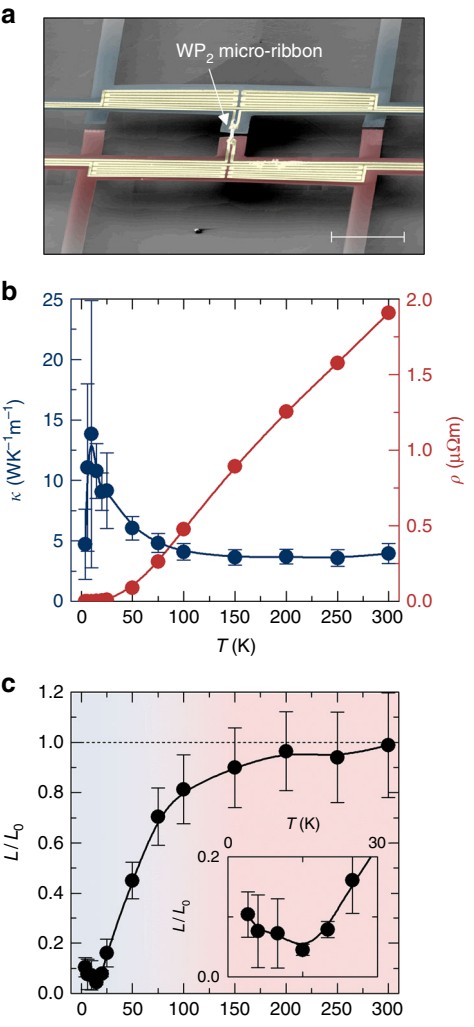

**Fig. 2** Violation of the Wiedemann–Franz law. **a** False-colored SEM image of a microdevice for measuring thermal transport that consist of two suspended platforms bridged by a 2.5-μm-wide WP₂ ribbon. The scale bar denotes 100 μm. **b** Thermal conductivity $\kappa$ (left axis) and electrical resistivity $\rho$ (right axis) of the micro-ribbon as a function of temperature. The error bars denote the error of the measurement as described in the Supporting Information. **c** Lorenz number $L = \kappa T \rho$, calculated from the data in **b** in units of the Sommerfeld value $L_0$. The error bars denote the error, coming from the thermal conductivity measurements. The inset shows a zoom of the low-temperature region

for both the sample as well as for the electrical contacts. Nevertheless, we calculate relatively large expected systematic errors as shown as error bars in the plot (see Supplementary Note 4). The experimentally extracted $\kappa$ as a function of $T$ is given in Fig. 2(b).

As shown in Fig. 2c, the Wiedemann–Franz relationship holds above 150 K, as the Lorenz number assumes the Sommerfeld value $L \approx L_0$. With decreasing temperature, however, $L$ is strongly reduced by more than one order of magnitude to a value below 0.05 $L_0$ at 20 K. We note, that our thermal conductance measurements are in agreement with independent measurements on macroscopic WP$_2$ crystals leading to $L \approx 0.25 L_0$ at $T = 11$ K, despite the full phonon contribution being included[32]. The interaction between the channel boundaries and the carriers should also influence the heat transport, because heat currents are not only relaxed by momentum conserving, but also by momentum-relaxing scattering events. This result implies, that either the phonon contribution is relatively small due to the metallic character of the WP$_2$, or that the electron-contribution is below $L_0$ even above 150 K. In any case the phonon thermal conductivity appears to be relatively small, which may be due to the heavy W-atoms that cause slow phonon velocities.

To interpret, we recall that in conventional conductors, the Wiedemann–Franz law holds, i.e., $L$ equals the Sommerfeld value $L_0 = \pi^2 \, k_B^2/(3e^2)$. Violations of the Wiedemann–Franz law typically are an indication of invalidity of the quasi-particle picture, strong difference between $\tau_{er}$ and $\tau_{mr}$ in quasi-particle scattering, or ambipolar physics. Phonon contributions to $\kappa$ enhance $L$. Conversely, small ($O(1)$) deviations below $L_0$ are usually observed in metals near 0.1 of the Debye temperature. Many ultra-pure metals at low temperatures, for example, can exhibit reduced values of $L/L_0 < 0.5$. Rhenium[33] and Silver[34] are among the most extreme cases with $L/L_0$ at and below 0.1. This effect is due to electron–phonon scattering and a transition from inelastic large angle to small-angle scattering processes[13,35]. In hydrodynamic systems, $L$ can become arbitrarily small because of the difference between the two relaxation times $\tau_{mr}$ and $\tau_{er}$ governing electrical and thermal transport, respectively[3,5,14,36]. Hence, we expect the ratio between $\tau_{mr}$ and $\tau_{er}$ in WP$_2$ to be at least one order of magnitude[5]. However, we note that this can only be a lower bound of the difference between the scattering times in the electron system because the residual phonon contributions are not subtracted from $\kappa$. The observed maximum in thermal conductance at around 10 K is consistent with this notion. Nevertheless, the consideration of a separate heat-conduction channel via the crystal lattice through phonons must be made with care in correlated materials. A recent claim argues that the hydrodynamic fluid may comprise both phonons and electrons and it is sometimes referred to as an electron–phonon soup[37]. In any case, the measured thermal conductance shown here includes both phonon and electron contributions.

The Lorenz value we obtained belongs to the lowest such values ever reported[30]. This is an indication for strong inelastic scattering, and, in combination with the charge transport data shown above, an independent evidence of a hydrodynamic electron fluid below 20 K.

## Magnetohydrodynamic transport.

Exploiting the well-justified conjecture of the hydrodynamic nature of transport in WP$_2$ at low temperatures, we can now manipulate and tune the viscosity of the electron fluid to obtain further information on the magnitude of $\tau_{mc}$. For this, the resistivity of the micro-ribbons is measured as a function of the magnetic field $\mathbf{B}$ at fixed $T$. $|\mathbf{B}| = B$ is set along the $b$-axis of the crystal and thus perpendicular to the direction of current flow. In an electron liquid, the viscosity is defined by the internal friction between layers of different

velocities[11], mediated by the exchange of electrons (Fig. 3a). The strength of the friction is given by the penetration depth $\lambda_p$ of the electrons, which is on the order of the mean free path between collisions, $\lambda_p \approx l_{mc}$, at zero magnetic field. When $\mathbf{B}$ is turned on, however, this penetration depth is limited by the cyclotron radius $r_c = m v_F/eB$. Thus, in a strong magnetic field, the viscosity should tend to zero, providing a mechanism for a large negative magneto-resistivity. Solving the corresponding magnetohydrodynamic equations results in the magnetic field-dependent viscosity along the $a$-axis of the crystal $\eta(B) = \eta_0/(1 + (2\tau_{mc}\omega_c)^2)$, where $\omega_c = v_F/r_c$ is the cyclotron frequency (see Supplementary Note 5 and Supplementary Figures 11–13 for details)[11,12,35].

As shown in Fig. 3b, we observe a large negative magneto-resistivity at low temperatures. When $\rho(B) - \rho_0$ is normalized by $w^{-2}$, all experimental data below 20 K collapse onto a single curve, matching the magnetic field-dependent viscosity model excellently. The average viscosity of the single traces at zero field $\eta_0$ equals the viscosity $\eta$ obtained from the $w^{-2}$ fits above. This agreement is an important cross-check, confirming our results

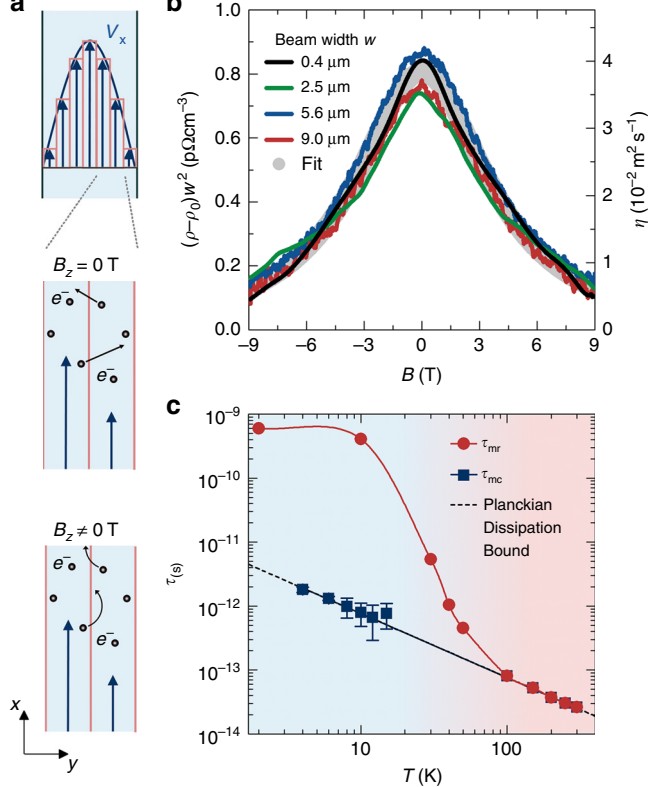

**Fig. 3** Magnetohydrodynamics and the Planckian bound of dissipation. **a** The origin of the decrease of the electron viscosity $\eta$ in a magnetic field $B$ (schematic) perpendicular to the current flow ($B = B_z$) and the sample width. The viscous friction between two adjacent layers of the electron fluid moving (the arrows point along the flow direction) with different velocities (length of the arrows) is determined by the depth of the interlayer penetration of the charge carriers $e^-$ (black dots). In a magnetic field, this depth is limited by the cyclotron radius. **b** $(\rho - \rho_0)/w^2$ as a function of $B$ for all four WP$_2$ ribbons investigated at 4 K (lines), where $\rho_0$ is the electrical resistance in zero field. The experimental data have been fitted by the magnetohydrodynamic model in the Navier–Stokes flow limit (gray dots). **c** Experimentally extracted momentum-relaxing and momentum-conserving relaxation times $\tau_{mr}$ and $\tau_{mc}$, respectively (symbols, with guide to the eye). The error bars denote the errors of the fits exemplarily displayed for 4 K in **a** (see Supplementary Information for details). The dashed line marks the Planckian bound on the dissipation time $\tau_h = \hbar/(k_B T)$

and the hydrodynamic interpretation. Furthermore, in accordance with the theory, $\eta(B)$ tends to zero as the magnetic field is enhanced.

**Extraction of the relaxation times**. The $\tau_{mc}$ extracted at individual $T$ values are plotted in Fig. 3c. Throughout the hydrodynamic regime, we find that the momentum-conserving scattering time in the electron liquid is tied to the Planckian limit[3,16,17] as $\tau_{mc} \sim \tau_\hbar = \hbar/(k_B T)$. As an important cross-check, we calculated the kinematic shear viscosity[11,12] from $\tau_{mc}$ as $\eta = v_F^2 \tau_{mc}/4$, and obtained consistent values with those extracted above (Supplementary Fig. 12), confirming the underlying dissipation bound.

For comparison, we also determined the timescale of the charge current relaxation related to the electrons' momentum. $\tau_{mr}$ is extracted from combined resistivity and Hall measurements on the bulk sample[28] (Supplementary Fig. 1a). With the carrier concentration $n$ (Supplementary Fig. 1b) obtained from Hall measurements and Shubnikov-de Haas oscillations[28], we calculated the mobility $\mu = (\rho e n)^{-1}$ (Supplementary Fig. 1c). Using the average effective mass of the charge carriers $m^* = 1.21 m_0$ ($m_0$ is the free-electron mass) and $\mu = e\tau_{mr}/m^*$, we eventually obtained $\tau_{mr}$ as a function of the temperature (Fig. 3c). At low temperatures, $\tau_{mr}$ is three orders of magnitudes larger than $\tau_{mc}$ and thus also than the Planckian bound. Consequently, $l_{mc} < w \ll l_{mr}$ (Supplementary Fig. 13), validating the hydrodynamic treatment of transport in WP$_2$ in this temperature regime.

While our experiments allow for the extraction of the relaxation times $\tau_{mr}$ and $\tau_{mc}$, the microscopic origin of the hydrodynamic transport regime remains elusive. However, recent ab initio calculations of the scattering time-resolved Fermi surfaces in WP$_2$ suggest that phonon related processes, rather than purely electron–electron processes, have a critical role in the emergence of the hydrodynamic behavior[38].

A natural question that arises from this observation is whether Planckian dissipation is exclusive to the hydrodynamic regime in WP$_2$. In the $T$-linear regime above 150 K, where electron–phonon Umklapp scattering yields $L = L_0$ (because $\tau_{mc} = \tau_{mr}$), we found that, despite their fundamental difference, electron–phonon Umklapp and electron-defect processes are also tied to $\tau_\hbar$. This observation indicates that the dissipation in WP$_2$ is generally tied to a characteristic timescale $\tau_\hbar$, regardless of the transport regime and the details of the underlying scattering mechanisms.

## Discussion

Our analysis suggests that WP$_2$ behaves like a typical Fermi Liquid in some respects and different in others. For example, the existence of quasi-particles is suggested by the Shubnikov-de Haas oscillations, weakly interacting particles are suggested by the large ratio between the dynamic viscosity and the number density (430 $\hbar$), and the degeneracy is implied from the ratio of the Fermi energy $E_F = 5.6$ eV (estimated from DFT calculations) and $k_B T$ at 4 K of about $1.7 \times 10^4$ (Supplementary Note 6 and Supplementary Figures 14–16 for details). However, a Fermi liquid is expected to have a $T^2$ scattering rate when clearly below the Debye temperature (for WP$_2$ estimated to be above 300 K from heat capacity measurements), which contradicts our results obtained at low temperatures. To find the relaxation time at the Planckian bound and linear in $T$ is unusual, but not in contradiction to the fundamental concepts of Fermi liquids[39].

The $T$-linearity observed in here could be entered onto the universal plot in of Bruin et al.[15], but all the other entries in that plot are either strongly interacting, close to quantum critical points or above the Debye temperature. How has WP$_2$ earned its right to participate in this universality?

In conclusion, our experiments strongly support the existence of a hydrodynamic electron fluid in WP$_2$. The accompanying independence of $\tau_{mc}$ and $\tau_{mr}$ allows the intrinsic thermal current relaxation process to be isolated, which is particularly elusive in other contexts. Remarkably, it turns out that the electron system in WP$_2$ generates entropy in a very simple and universal way in which the only relevant scale is the temperature.

## Methods

**WP$_2$ single-crystal growth of the bulk sample**. Crystals of WP$_2$ were prepared by chemical vapor transport method. Starting materials were red phosphorous (Alfa-Aesar, 99.999%) and tungsten trioxide (Alfa-Aesar, 99.998%) with iodine as a transport agent. The materials were taken in an evacuated fused silica ampoule. The transport reaction was carried out in a two-zone furnace with a temperature gradient of 1000 to 900 °C for serval weeks. After reaction, the ampoule was removed from the furnace and quenched in water. The metallic-needle crystals were characterized by X-ray diffraction.

**Electrical transport measurements on the bulk sample**. The electrical resistance of the bulk sample is determined within a four-probe configuration under isothermal conditions with an AC bias current of r.m.s. 3 mA at 93.0 Hz, using standard lock-in technique. As for the microbeams, the current is applied along the $a$-axis of the crystal. Magnetoresistance measurements are performed with a magnetic field applied perpendicular to the direction of current flow, within the $b$–$c$ plane. The Hall resistance is measured under the same conditions with a magnetic field applied along the $b$-axis. More details on the bulk characterization can be found in Kumar et al.[28] All measurements are carried out in a temperature-variable cryostat, equipped with a superconducting ±9 T magnet (PPMS Dynacool).

**Electrical transport measurements on the microbeams**. All transport measurements are performed in a temperature-variable cryostat (Dynacool, Quantum Design) in vacuum. The cryostat is equipped with a ± 9 T superconducting magnet, swept with a rate of 5 mT s$^{-1}$. After fabrication, the microbeam devices are mounted on a sample holder and wire bonded. Electrical resistance measurements on the microbeams are carried out in a quasi-four-probe configuration under isothermal conditions with AC bias currents of r.m.s. 100 µA max at 6.1 Hz, using standard lock-in technique. The quasi-four-terminal resistivity measurements exclude the resistance of the contact lines, but can in principle include interface resistances at the metal/semimetal junctions. However, we find no evidence for any resistance contribution from this interface (see Supplementary Note 2 for details).

**Fabrication of the MEMS platforms for heat transport measurements**. The fabrication process of the MEMS devices (Supplementary Fig. 8) is similar to the process of Karg et al.[31] A silicon wafer was coated with a 150 nm-thick layer of low-stress silicon nitride (SiMat, Germany). Gold lines with Cr adhesion layer were patterned using optical lithography, metal evaporation and a standard lift-off process. The under-etched regions were defined using optical lithography and etching of the silicon nitride. Finally, the devices were released using wet etching and critical-point drying.

The devices consist of two MEMS platforms each carrying a four-probe resistor of 430 and 700 Ω, respectively, and two electrical leads to contact the sample, such that the WP$_2$ sample can be measured electrically using a four-probe geometry. Each platform is connected to the wafer via four 1.2-mm-long silicon nitride suspension legs, two of which carry three gold leads each. Two nitride bridges connecting the two platforms served to stabilize the device while placing the sample and were cut using a focussed ion beam before the measurement. The thermal conductance of each platform was $1.6 \times 10^{-5}$ W K$^{-1}$ at room temperature and showed the expected slight increase in conductance at lower temperatures caused by the temperature-dependent thermal conductivity of silicon nitride and gold. The dimensions of the platform lead to a thermal conductance of the coupling between each heater/sensor platform to the chip carrier of $1.6 \times 10^{-5}$ W K$^{-1}$ at room temperature. This value is 50 times larger than the sample thermal conductance of the WP$_2$ sample of $3 \times 10^{-7}$ W K$^{-1}$ at room temperature.

**Sample mounting on MEMS platform**. The sample was mounted using a micromanipulator under an optical microscope. After positioning the sample, electrical contacts were deposited using electron-beam induced metal (Pt) deposition. Care was taken not to expose the WP$_2$ sample to the electron beam at any point in time except in the contact region during metal deposition. After successful mounting, the WP$_2$ sample bridges the gap between the two platforms, each equipped with a heater/sensor and two electrical leads to the WP$_2$ sample. We note that in favor of large electrical contacts and thereby good thermal coupling, the pairs of electrodes at either end of the WP$_2$ sample are not separated to allow subtracting the electrical contact resistance. However, the significant electrical resistance of the long microfabricated leads can be determined and subtracted from the measurement. Given the electrical conductance observed, we estimate a potential systematic error due to electrical contact resistance to be within the scatter of the experiments.

**Thermal transport measurements**. The method employed in this study is based on the method developed by Li Shi et al.[40], and refined since two metal resistors are fabricated using lithography. They serve as both micro-heaters and thermometers. The resistors are read out using the four-probe technique and are fabricated on structured silicon nitride membranes to avoid thermal cross-talk via heat conduction through the substrate. The measurements are performed in vacuum to avoid thermal cross-talk through air conduction.

In detail, we proceeded as follows. First, the temperature calibration step was performed. For this, the sample was settled to set temperatures between 4 and 300 K. The resistor on each platform was characterized measuring the voltage drop for a given current for currents between $-5$ and $5 \times 10^{-4}$ A in increments of 2 μA, in DC, from 5 to 40 mA, in AC using phase-sensitive detection modulated at 6 Hz. The electrical resistance extrapolated to the limit of zero current was used for calibration (Supplementary Fig. 10). The values measured using DC and AC operation match within the measurement error, confirming the correct choice of modulation frequency. The resulting resistance versus temperature plot shown in Supplementary Fig. 9 is used for temperature calibration of the two sensors operated as resistive thermometers.

For the subsequent transport measurements, polynomial fits for the resistance versus temperature for the thermometers were used; for temperatures above 50 K, we use a 2nd-order polynomial fit, for the lower temperatures we use a 5th-order polynomial fit. At lower temperatures, the resistance versus temperature plot goes through a minimum at around 10.4 K. This is the expected behavior of gold at these temperatures[41]. As a consequence, the sensitivity is small at around this temperature and, consequently the systematic error is larger compared to high temperatures. The error bars plotted in Fig. 2b of the main manuscript comprise this variation in sensitivity, as well as known systematic errors and measurement uncertainties.

After the temperature calibration, heat transport measurements were performed. At each cryostat temperature, a small sensing current of 10 nA was applied to heater/sensor 2, while the current was ramped as described above in heater/sensor 1. At all times, the current through the $WP_2$ sample was zero and the leads floating. Both sides were probed using phase-sensitive detection. For the analysis, we follow the method described by Shi et al.[40] and refined by Karg et al.[31] An example is given in Supplementary Fig. 10 for the cryostat temperature of 100 K. In summary, for each current the electrical resistance in the heater and the long leads on the suspension legs is determined. Then, using the current and the resistances, the effective Joule heating power $P_{eff}$ is calculated. From the measured resistance values $R_{Heater1}$ and $R_{Heater2}$, the temperature rise, $\Delta T_1$ and $\Delta T_2$ of both heater/sensors is calculated using the temperature calibration (Supplementary Fig. 9). The thermal conductance of the MEMS device and the $WP_2$ sample is then extracted from the slope of plots like Supplementary Fig. 10c and d. The same results were obtained using the symmetry check at one temperature where the roles of heater and sensor side was swapped, which reconfirms the treatment of the two sensors with different resistances.

The uncertainty of the thermal conductance measurements is calculated considering known systematic errors and measurements uncertainties, the largest contributions coming from the effects of device and sample geometry, electrical resistance measurement of the sensors, and the fitting procedure. The sensitivity of electrical resistance of the metal thermometers is strongly reduced at low temperatures. At low temperatures, this is the main contribution to the reported error (see the Supplementary Note 4 for a detailed discussion).

## Data availability

The data that support the findings of this study are available from the corresponding author upon request.

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

## Acknowledgements

We thank Steffen Reith and Valentina Troncale for technical support, as well as Stuart Parkin, Philip Moll, Subir Sachdev, Sean Hartnoll, Jan Zaanen, and Andrew Mackenzie for fruitful discussions. We also acknowledge support by Walter Riess and Kirsten Moselund, and thank Charlotte Bolliger for copy-editing. Fabian Menges gratefully acknowledges the support from the Society in Science—The Branco Weiss Fellowship and a Swiss National Science Foundation postdoctoral fellowship.

## Author contributions

J.G., F.M., C.F., and B.G. conceived the experiment. C.S., N.K., and V.S. synthesized the single-crystal bulk samples. U.D. fabricated the suspended platforms. J.G., F.M., and B.G. produced the micro-ribbons and processed the devices. J.G. carried out the transport measurements on the micro-ribbons with the help of R.Z. J.G. and B.G. analyzed the data. C.S. and N.K. performed the Hall and resistivity measurements on the bulk samples. Y.S. calculated the band structure. B.G. supervised the project. All authors contributed to the interpretation of the data and to the writing of the manuscript.

## Additional information

**Competing interests:** The authors declare no competing interests.

