## [Peer Review File · Nature Communications]

Reviewers' comments:

Reviewer #1 (Remarks to the Author):

In this paper the authors argue that they have found experimental signatures of hydrodynamic electron flow in WP2 at low temperatures. This is indeed an interesting point for high mobility topological materials. However, I found that the paper is not convincing enough. Below are some questions.

- 1) Before the discussion of probable signatures of hydrodynamics in WP2, what is the signature of electron-electron correlations, which the authors argued, in this material?
- 2) Because the momentum relaxed mean free path of electrons is much longer than the channel width in this material, it is natural to expect a w dependent resistivity at low temperatures, see Fig. 1b. A ballistic transport in constricted channel may provide a simple explanation to these data.
- 3) As the phonon contribution is included to the thermal conductivity, it is strange that L goes to L_0 above 150 K, see Fig. 2c. In other words, the phonon heat conductance is missing. Why?
- 4) P10, "We note, that our thermal conductivity measurements are in excellent agreement with independent measurements on macroscopic WP2 crystals leading to $L = 0.1 L_0$ at $T = 15$ K.....". Why is the thermal conductance in the hydrodynamic regime similar to that in a macroscopic system, where hydrodynamics is not expected?
- 5) P10, "Ultra-pure metals at low temperatures, for example, can reach of $L/L_0 \sim 0.1$ in extreme cases." Here, the readers would like to know the exact materials and measurement conditions, proper references should be provided.
- 6) Thermal conductivity was measured for only one sample (the 2.5- μm one), the authors should mention why others were not measured. It is clear that these data, if available, are very informative
- 7) Ref. 2, the citation information is missing.

Reviewer #2 (Remarks to the Author):

In their manuscript NCOMMS-18-13710, the authors provide evidence of hydrodynamic transport in WP2. In particular, using narrow micro-ribbons, they observe a low-temperature resistivity proportional to w^{-2} , where w is the transverse width of the ribbon, consistent with the Gurzhi effect. They also observe a modest enhancement of the thermal conductivity, along with a strong enhancement of the low-temperature conductivity (leading to a strong suppression of the Lorenz ratio). Finally, they observe negative magnetoresistance, which they interpret as a suppression of the viscosity.

All of these results do indeed suggest hydrodynamic transport, and therefore I am inclined to recommend this paper for publication. There are however some key questions that I feel the authors must address before I can do so. In addition, I believe the manuscript itself would benefit from a partial shift of focus.

Much of the main text is devoted to the 'Planckian bound' for the inelastic scattering rate (which the authors call " $1/\tau_{mc}$ "). This bound comes from attempts by theorists to understand strange metal behavior in certain strongly correlated systems using AdS/CFT (really "AdS/CMT") duality, and is not by any means universally accepted across the community. Although the authors carefully distinguish the momentum relaxation and energy relaxation/inelastic scattering rates, they seem to conflate hydrodynamics in electron systems with these somewhat speculative ideas when in fact, a Fermi liquid can be perfectly hydrodynamic.

The authors themselves admit at the end of the main text that WP2 is likely a good Fermi liquid,

and the data supports this: large low-temperature mobility, large carrier density, etc. A Fermi liquid can be hydrodynamic: all that is required is that the inelastic scattering rate due to electron-electron interactions is the fastest scale in the problem, faster than electron-impurity, electron-phonon, etc. In typical metals and semiconductors, it is hard to achieve the hydrodynamic regime owing to the large value of E_F (which suppresses the inelastic rate) and the finite density of impurities. From what I can tell, the data here seems to support the idea that these WP2 microribbons are sufficiently clean that at low temperatures they behave hydrodynamically.

The hydrodynamics of a Fermi liquid can be entirely understood via the kinetic equation, with microscopic collision integrals computed directly from Fermi's golden rule. Such an analysis was carried out for graphene in order to understand a violation of the Mott relation observed very recently, see Xie and Foster, PRB 93, 195103 (2016). There is no need to invoke fancy ideas from string theory in this case.

But this leads to a key question with respect to the present manuscript. The authors extract the inelastic scattering rate from the magnetoresistance, and this seems to saturate the Planckian bound. This is at odds with Fermi liquid theory for a degenerate system, which would predict $1/\tau_{in} \sim (k_B T)^2/E_F$.

Moreover, the absolute value of the measured viscosity divided by the carrier density is 430 times \hbar . For a system saturating the Planckian bound, one expects all scales to be determined by temperature alone (quantum critical behavior). Then the viscosity is roughly the thermally-activated carrier density. Since the latter should be much less than the total carrier density in the degenerate regime, the large value obtained here (430) suggests that the physical quasiparticle scattering rate is NOT given by $k_B T / \hbar$.

The authors must address this issue. More space should be devoted in the main manuscript to the Fermi liquid character (or not) of the samples at low temperature. I.e., I could not find the Fermi energy anywhere in the manuscript. What is the ratio of $k_B T$ to E_F for the relevant temperature range?? This should be the first number given in a discussion of electronic hydrodynamics!

The mismatch between the absolute value of the viscosity and its bound versus the apparently Planckian rate derived from the magnetoresistance suggests to me that the timescale in the latter might not be the quasiparticle decay rate. But then this calls into question the relevance of the Planckian bound here (the presence or absence of which I do not think is necessary to warrant publication of the main experimental findings).

Since the system is a Weyl semimetal, is it clear that there isn't another source for the magnetoresistance (chiral anomaly, etc)?

Another minor issue is that the authors appear to conflate having a large value of $k_F l_{el}$ (where l_{el} is the elastic scattering length due to impurity scattering) with Fermi liquid theory, as opposed I presume to the "strong correlated soup" that is envisioned by the AdS/CFT studies. One should remember however that $k_F l_{el}$ really tells you how good of a diffusive Fermi liquid you have, and doesn't directly imply anything about correlations. Diffusive Fermi liquids however satisfy the Mott and Wiedemann-Franz relations.

In my view if the authors can clarify the Fermi liquid character of the sample, and discuss the discrepancy of the inelastic rate extracted from the magnetoresistance to the expected Fermi liquid lifetime, then I will be apt to recommend this paper for publication.

Finally, it might round out the AdS/CMT-heavy references to cite a few of the earlier works on electron hydrodynamics, e.g.

R. N. Gurzhi

Journal of Experimental and Theoretical Physics 17 521 (1963)

and on thermoelectric and magnetotransport in the hydrodynamic regime for graphene:

Mueller and Sachdev, PRB 78, 115419 (2008)

Foster and Aleiner, PRB 79, 085415 (2009)

Mueller, Schmalian, and Fritz, PRL 103, 025301 (2009)

Response to Reviewers

We would like to thank both reviewers for their review. The time they took to describe the issues make this an enjoyable scientific discussion. We hope to discuss with them once in person. We have found the comments valuable and made a number of changes to clarify and /or answer the questions.

Reviewer 1

The reviewer writes:

In this paper the authors argue that they have found experimental signatures of hydrodynamic electron flow in WP₂ at low temperatures. This is indeed an interesting point for high mobility topological materials. However, I found that the paper is not convincing enough. Below are some questions

Response:

We are grateful to the referee for the detailed questions. We have revised our work along the lines of these questions, which we consider having significantly improved the manuscript. The new version clarifies on the interaction mechanism, the connection to ballistic transport, the phonon contribution to the heat transport and the Lorenz number. Below, we address in detail the specific comments included in the report.

The reviewer writes:

1) Before the discussion of probable signatures of hydrodynamics in WP₂, what is the signature of electron-electron correlations, which the authors argued, in this material?

Response:

We thank the reviewer for this comment. In the previous version of the manuscript we were not sufficiently precise regarding the microscopic origin of the hydrodynamic transport regime in WP₂. In fact, our experiments go only as far as to extract the relaxation times. The microscopic scattering mechanisms are neither extracted nor claimed. They could be electron-electron as well as small-angle electron-phonon scattering. We note, however, that the origin of the hydrodynamic transport in WP₂ has very recently been theoretically discussed in a preprint, claiming an electron-phonon mechanism (<https://arxiv.org/abs/1804.06310>). For clarification, we have now added an explicit statement in the main text that our experiment does not allow for a conclusion on the microscopic origin of the hydrodynamic behaviour in WP₂:

“While our experiments allow for the extraction of the relaxation times τ_{nr} and τ_{mc} , the microscopic origin of the hydrodynamic transport regime remains elusive. However, recent ab-initio calculations of the scattering time-resolved Fermi surfaces in WP₂ suggest that phonon related processes, rather than purely electron-electron processes, play a critical role in the emergence of the hydrodynamic behavior.³⁸”

We added the corresponding reference. To avoid confusion from the start, we also changed the first sentence of the abstract to:

“Materials in which electrons strongly interact with each other or with phonons exhibit interesting phenomena such as metal-insulator transitions and high-temperature superconductivity.”

As explained in detail in the answers to referee 2, a lot of observations just point at an “ordinary” Fermi liquid in WP₂. This is actually one of the interesting aspects of our findings.

The reviewer writes:

2) Because the momentum relaxed mean free path of electrons is much longer than the channel width in this material, it is natural to expect a w dependent resistivity at low temperatures, see Fig. 1b. A ballistic transport in constricted channel may provide a simple explanation to these data.

Response:

We fully agree with the reviewer that a ballistic non-specular surface scattering could also cause a width-dependence of the resistivity when the specimen thickness is less than the electron mean free path. As we explain in the main text, there are now two physical regimes that can be distinguished by the power-law of the width dependence: A power β of $0 > \beta \geq -1$ indicates ballistic conduction in the sample, which means that the momentum relaxing mean free path equals the thermal current relaxing mean free path. A power β of $-1 > \beta \geq -2$ indicates hydrodynamic conduction in the sample, which means that the momentum relaxing mean free path is much longer than the thermal current relaxing mean free path. The lower limits of -1 and -2 for the ballistic and hydrodynamic case, respectively, occur in the limit of fully diffusive boundary scattering. Because we observe $\beta \approx -2$ at low temperatures, the effects cannot be explained by ballistic transport. For clarification, we have strengthened the related discussion of the manuscript in the following way:

“The exponent β characterizes different transport regimes. In the well-established ballistic regime ($w \ll l_{er}, l_{mr}$), for example, a power of $0 > \beta \geq -1$ occurs and the electrical resistivity in the limit of fully diffusive scattering is given by $\rho \sim w^{-1}$. Further, in a hydrodynamic fluid ($l_{er} \ll w \ll l_{mr}$), the flow resistance is determined solely by the interaction with the sample boundaries, reducing the average flow velocity of the electron fluid (Fig. 1 (c)). As a consequence, a power of $-1 > \beta \geq -2$ is indicative for hydrodynamic transport.”

In addition we note that the observed violation of the Wiedemann-Franz law in this temperature regime provides independent evidence for the hydrodynamic flow. In the ballistic regime, the WF law should explicitly hold, because the relaxation times for thermal and electrical currents are both conserved. As explained in the main text, in the hydrodynamic regime, the WF law must be violated.

The reviewer writes:

3) As the phonon contribution is included to the thermal conductivity, it is strange that L goes to L_0 above 150 K, see Fig. 2c. In other words, the phonon heat conductance is missing. Why?

Response:

We are not aware of any directly applicable experimental method to differentiate between charge and lattice contribution to heat conduction in our sample. We point out that WP₂ behaves like a metal. In metals, oftentimes the phonon contribution is assumed to be independent from the electronic contribution. Wiedemann-Franz is used to estimate the electron contribution to heat transport, usually dominating

strongly. However, when the electronic contribution reduces via breaking Wiedemann Franz law, we need to expect that the remaining thermal conductivity may have a larger phonon contribution. In our case, the contribution of phonon would have to be within the experimental uncertainty of ~20%. Therefore the result implies, that either the phonon contribution is relatively small, or that the electron-contribution is below L_0 even above 150K. In any case the phonon thermal conductivity appears to be relatively small. This we found plausible because of the heavy W that may cause slow phonon velocities. We realize that the lack of detail in the discussion of phonon contributions may render the paper inconclusive. We added to the original description in the main text:

“This result implies, that either the phonon contribution is relatively small due to the metallic character of the WP₂, or that the electron-contribution is below L_0 even above 150 K. In any case the phonon thermal conductivity appears to be relatively small, which may be due to the heavy W-atoms that cause slow phonon velocities.”

The reviewer writes:

4) P10, “We note, that our thermal conductivity measurements are in excellent agreement with independent measurements on macroscopic WP₂ crystals leading to $L = 0.1 L_0$ at $T = 15$ K.....”. Why is the thermal conductance in the hydrodynamic regime similar to that in a macroscopic system, where hydrodynamics is not expected?

Response:

The reviewer appears to imply that the thermal conductivity (thermal conductance normalized by dimensions) may well be different between a macroscopic (mm-sized) and microscopic (micron sized) sample. This is in a sense what we see. We report the same Lorenz ratio for two sample sizes for two samples with very different electrical conductivity. Therefore the thermal conductance (or effective thermal conductivity) is also different in the two samples. For us it is not unexpected that the reduction of electrical conductance due to reduced channel width should result in a similar reduction on thermal conductivity. In a hand-waving sense, one may apply a similar rationale like the difference between convection and conduction in ordinary gases and liquids. While convective transport would be reduced in small channel, the conduction should not. The transport in WP₂ is therefore similar to convection.

The interaction between the channel boundaries and the carriers should also influence the heat transport, because heat currents are not only relaxed by momentum conserving, but also by momentum relaxing scattering events. A microscopic picture for this is drafted already by Ghurzi et al., JETP 1989. We added the following explanatory sentence to the manuscript:

“The interaction between the channel boundaries and the carriers should also influence the heat transport, because heat currents are not only relaxed by momentum conserving, but also by momentum relaxing scattering events”

The reviewer writes:

5) P10, “Ultra-pure metals at low temperatures, for example, can reach of $L/L_0 \sim 0.1$ in extreme cases.” Here, the readers would like to know the exact materials and measurement conditions, proper references should be provided.

Response:

We added the new wording:

“Many ultra-pure ordinary metals at low temperatures, for example, can exhibit reduced values of $L/L_0 < 0.5$, Rhenium [Hust] and Silver [Glees] are among the most extreme cases with L/L_0 at and below 0.1.”

And the corresponding references:

K. Gloos, C. Mitschka, F. Pobell and P. Smeibidl, Cryogenics 30, 1990, 14-18.

J. G. Hust and L. L. Sparks, NBS Technical Note 634 (1973).

The reviewer writes:

6) Thermal conductivity was measured for only one sample (the 2.5-um one), the authors should mention why others were not measured. It is clear that these data, if available, are very informative

Response:

Yes, we agree that more samples would be informative. The reasons for not (yet) measuring more are mainly budget and time constraints. The thermal conductance measurements are the most costly in terms of sample fabrication (manual placement on MEMS platforms) and measurement time (all the calibration takes a lot of measurement time at low temperatures) and corresponding budget.

We have therefore made extra effort in order to provide credible data in such that we made a particular conservative error estimate. In parallel, we have sent a macroscopic bulk crystal (mm-size) to Kamran Behnia's group in Paris for an independent cross-check of our obtained Lorenz ratio. As explained above, these experiments reproduce this part of our results. We have cited them accordingly in our manuscript.

The reviewer writes:

7) Ref. 2, the citation information is missing.

Response:

We thank the reviewer for pointing that out. We apologize and have corrected the citation information.

Reviewer 2

The reviewer writes:

In their manuscript NCOMMS-18-13710, the authors provide evidence of hydrodynamic transport in WP2. In particular, using narrow micro-ribbons, they observe a low-temperature resistivity proportional to w^{-2} , where w is the transverse width of the ribbon, consistent with the Gurzhi effect. They also observe a modest enhancement of the thermal conductivity, along with a strong enhancement of the low-temperature conductivity (leading to a strong suppression of the Lorenz ratio). Finally, they observe negative magnetoresistance, which they interpret as a suppression of the viscosity.

All of these results do indeed suggest hydrodynamic transport, and therefore I am inclined to recommend this paper for publication. There are however some key questions that I feel the authors must address before I can do so. In addition, I believe the manuscript itself would benefit from a partial shift of focus.

Response:

We thank the reviewer for the careful assessment of our work and are delighted that the reviewer is “*inclined to recommend this paper for publication.*” We have now incorporated all the suggestions which we address separately below.

The reviewer writes:

Much of the main text is devoted to the "Planckian bound" for the inelastic scattering rate (which the authors call " $1/\tau_{mc}$ "). This bound comes from attempts by theorists to understand strange metal behavior in certain strongly correlated systems using AdS/CFT (really "AdS/CMT") duality, and is not by any means universally accepted across the community. Although the authors carefully distinguish the momentum relaxation and energy relaxation/inelastic scattering rates, they seem to conflate hydrodynamics in electron systems with these somewhat speculative ideas when in fact, a Fermi liquid can be perfectly hydrodynamic.

Response:

We understand the reviewer’s concern. We agree that the established concepts of a Fermi liquid and the more speculative aspects of AdS/CFT approaches must be separated more clearly in the wording of the manuscript.

Maybe the most important change in this sense is the notion that the “Planckian bound” does not strictly require the notion of AdS/CFT. In fact, in one of the earliest papers on hydrodynamics in a Fermi liquid it is stated decades before the AdS/CFT predictions:

“However, there is another condition which limits the range of application of the theory to much lower temperatures. This is that the excitation energies, which are of the order T , must be considerably greater than the quantum indeterminacy in the energy which is due to collisions, i.e.

$$\tau \gg \hbar/T. \quad (8.12)$$

where τ is the time between collisions. We note that the condition (8.12) must be fulfilled not just for the calculation of the kinetic coefficients, but also for the whole theory of a Fermi liquid to be valid.” (The theory of a fermi liquid, A A Abrikosov and I M Khalatnikov 1959 Rep. Prog. Phys. 22, 329).

The following changes were made to the manuscript to address this point:

“Despite the significant difference in the microscopic mechanisms behind momentum- and energy-current-relaxing collisions, both processes should be limited by the quantum indeterminacy in the energy dissipation with a time scale larger than $\tau_{\hbar} = \hbar/(k_B T)$, where τ_{\hbar} is determined only by the Boltzmann constant k_B , the reduced Planck constant \hbar and the temperature T . This concept of, sometimes called, “Planckian dissipation” follows directly from the uncertainty principle when one applies equipartition of energy and any degree of freedom only carries $k_B T$.”

“However, whether the AdS/CFT predictions prove to be relevant and provide a true bound for hydrodynamic electron systems is still an open question.”

The reviewer writes:

The authors themselves admit at the end of the main text that WP2 is likely a good Fermi liquid, and the data supports this: large low-temperature mobility, large carrier density, etc. A Fermi liquid can be hydrodynamic: all that is required is that the inelastic scattering rate due to electron-electron interactions is the fastest scale in the problem, faster than electron-impurity, electron-phonon, etc. In typical metals and semiconductors, it is hard to achieve the hydrodynamic regime owing to the large value of E_F (which suppresses the inelastic rate) and the finite density of impurities. From what I can tell, the data here seems to support the idea that these WP2 microribbons are sufficiently clean that at low temperatures they behave hydrodynamically.

Response:

Yes, this is one of the main conclusions from our paper.

The reviewer writes:

The hydrodynamics of a Fermi liquid can be entirely understood via the kinetic equation, with microscopic collision integrals computed directly from Fermi's golden rule. Such an analysis was carried out for graphene in order to understand a violation of the Mott relation observed very recently, see Xie and Foster, PRB 93, 195103 (2016). There is no need to invoke fancy ideas from string theory in this case.

Response:

Again, we agree that hydrodynamics is not per se novel and does not require fancy ideas (the referee probably means invoking the AdS/CFT predictions). As a side remark, the violation of the Mott relation in

graphene is a little different. We considered at some point to discuss it in our paper and found it too large a stretch, because the argumentation in the end is rather different from what can be applied to WP₂. The kinetic equation alone, however, does not suffice to explain our findings. In our case the uncertainty principle must also be invoked. This has been claimed before for superconductors and some metals (*Bruin et al., Science 339, 804 (2013)*), i. e. for systems for which AdS/CFT is being considered and for systems for which it is not being considered.

The reviewer writes:

But this leads to a key question with respect to the present manuscript. The authors extract the inelastic scattering rate from the magnetoresistance, and this seems to saturate the Planckian bound. This is at odds with Fermi liquid theory for a degenerate system, which would predict $1/\tau_{in} \sim (k_B T)^2/E_F$.

Response:

We fully share the surprise and are similarly amazed by the T -linear scattering rate observed in WP₂, because conventional Fermi liquid theory is typically not described using this bound. As stated above, this additional criterion was mentioned by Abrikosov and Khalatnikov (1959). However, we have done three independent measurements to verify this result (width-dependence, thermal transport and magnetic-field dependence), which are all fully consistent. In fact, we have now performed electric and thermal measurements on various semi-metallic compounds, showing similar results. So far, the microscopic picture is indeed puzzling.

The reviewer writes:

Moreover, the absolute value of the measured viscosity divided by the carrier density is 430 times \hbar . For a system saturating the Planckian bound, one expects all scales to be determined by temperature alone (quantum critical behavior). Then the viscosity is roughly the thermally-activated carrier density. Since the latter should be much less than the total carrier density in the degenerate regime, the large value obtained here (430) suggests that the physical quasiparticle scattering rate is NOT given by $k_B T / \hbar$. The authors must address this issue. More space should be devoted in the main manuscript to the Fermi liquid character (or not) of the samples at low temperature.

Response:

Yes, we fully agree with the reviewer and note again that, so far, the microscopic picture is puzzling. To highlight the conflicting experimental observations, we have now followed the reviewer's advice and added a deeper discussion about the Fermi-liquid behaviour in WP₂ before the conclusion section in the manuscript.

“Our analysis suggests that WP₂ behaves like a typical Fermi Liquid in some respects and different in others. For example, the existence of quasiparticles is suggested by the Shubnikov-de Haas oscillations, weakly interacting particles are suggested by the large ratio between the dynamic viscosity and the number density (430 \hbar), and the degeneracy is implied from the ratio of the Fermi energy $E_F = 5.6$ eV (estimated

from DFT calculations) and $k_B T$ at 4 K of about 1.7×10^4 . However, a Fermi liquid is expected to have a T^2 scattering rate when clearly below the Debye temperature (for WP_2 estimated to be above 300 K from heat capacity measurements), which contradicts our results obtained at low temperatures. To find the relaxation time at the Planckian bound and linear in T is unusual but not in contradiction to the fundamental concepts of Fermi liquids [Abrikosov 1959].

The reviewer writes:

I.e., I could not find the Fermi energy anywhere in the manuscript.

Response:

The Fermi energy is 5.6 eV (from to DFT calculations).

The reviewer writes:

What is the ratio of $k_B T$ to E_F for the relevant temperature range?? This should be the first number given in a discussion of electronic hydrodynamics!

Response:

At 4 K, the ratio of $k_B T$ to E_F is 6.1×10^{-5} . This ratio has now been added to the Fermi liquid discussion.

The reviewer writes:

The mismatch between the absolute value of the viscosity and its bound versus the apparently Planckian rate derived from the magnetoresistance suggests to me that the timescale in the latter might not be the quasiparticle decay rate. But then this calls into question the relevance of the Planckian bound here (the presence or absence of which I do not think is necessary to warrant publication of the main experimental findings).

Response:

Yes, indeed. The role of the Planckian time scale is not understood in WP_2 . However, the data stands as it is. To highlight this question, we have now directly phrased it at the end of the manuscript:

“The T -linearity observed in here could be entered onto the universal plot in of Bruin et al.,⁴ but all the other entries in that plot are either strongly interacting, close to quantum critical points or above the Debye temperature. How has WP_2 earned its right to participate in this universality?”

The reviewer writes:

Since the system is a Weyl semimetal, is it clear that there isn't another source for the magnetoresistance (chiral anomaly, etc)?

Response:

No, the negative magnetoresistance induced by the chiral anomaly in Weyl semimetals only appears in magnetic fields that are aligned in parallel to the direction of current flow. However, the magnetic field in our experiments is always applied perpendicular.

The reviewer writes:

Another minor issue is that the authors appear to conflate having a large value of $k_F l_{el}$ (where l_{el} is the elastic scattering length due to impurity scattering) with Fermi liquid theory, as opposed I presume to the "strong correlated soup" that is envisioned by the AdS/CFT studies. One should remember however that $k_F l_{el}$ really tells you how good of a _diffusive_ Fermi liquid you have, and doesn't directly imply anything about correlations. Diffusive Fermi liquids however satisfy the Mott and Wiedemann-Franz relations.

Response:

We have given the ratio because it has been used to distinguish the "correlated soup" from Fermi liquids using the so-called Mott-Ioffe rule. The conclusion is that we can describe the system using quasi-particles. And yes, we agree, this does not yet say anything about correlations. To make this clearer we added at the corresponding section

"This does not directly imply anything about correlations."

The reviewer writes:

Finally, it might round out the AdS/CMT-heavy references to cite a few of the earlier works on electron hydrodynamics, e.g.

*R. N. Gurzhi
Journal of Experimental and
Theoretical Physics 17 521 (1963)*

and on thermoelectric and magnetotransport in the hydrodynamic regime for graphene:

*Mueller and Sachdev, PRB 78, 115419 (2008)
Foster and Aleiner, PRB 79, 085415 (2009)
Mueller, Schmalian, and Fritz, PRL 103, 025301 (2009)*

Response:

We thank the reviewer again for her or his valuable comments. We have now added the suggested references and hope our work will nourish further understanding of the interesting puzzles of hydrodynamic electron transport and the T -linear scattering rate in metals.

REVIEWERS' COMMENTS:

Reviewer #1 (Remarks to the Author):

The authors have addressed most of my concerns in the revised version. The only point not really clarified is question 4. They pointed out that the reduced L/L_0 found in this work, as a signature of hydrodynamic transport, is in agreement with the one found for macroscopic samples. Does this mean the hydrodynamics is sample-dimension independent? The readers want to know the physical significance of this agreement. Otherwise, I would evaluate the effort of the authors and support acceptance of the paper.

Reviewer #2 (Remarks to the Author):

The authors have conceded all of my points, except my misunderstanding of the magnetoresistance.

The remaining picture is quite puzzling, however, since all indications are that the system is a good Fermi liquid that is DEEP in the degenerate regime. The statement about uncertainty prescribing the constraint $\tau > 1/k_B T$ does not contain any more or less information than the proposed "Planckian bound," and certainly does not explain why such a time scale can arise in a system that is otherwise well-captured by Fermi liquid theory.

Despite these reservations, my overall inclination is that these results are interesting, and probably worth publishing in Nature Communications, so long as the Editors are aware that these results raise more (potentially very important) questions than they answer.

Response to Reviewers

We are delighted that both reviewers recommend our paper for publication and thank both reviewers again for their valuable comments and suggestions. In the following, please find our comments to their final remarks.

Reviewer 1

The reviewer writes:

The authors have addressed most of my concerns in the revised version. The only point not really clarified is question 4. They pointed out that the reduced L/L_0 found in this work, as a signature of hydrodynamic transport, is in agreement with the one found for macroscopic samples. Does this mean the hydrodynamics is sample-dimension independent? The readers want to know the physical significance of this agreement. Otherwise, I would evaluate the effort of the authors and support acceptance of the paper.

Response:

The hydrodynamics itself is indeed independent of the sample size. However, its signature in transport coefficients through contributions of the viscosity should not be independent. The effect of the viscosity on the electrical resistivity should be more pronounced as on the thermal conductivity in the finite carrier density regime and vice versa in the zero-density regime. Within our measurement precision, defined by the error of the thermal conductivity measurements (as discussed in detail in the Supplementary Information), we cannot resolve any difference. A size-dependent thermal conductivity study would be certainly interesting and is on our future measurement agenda, but requires further years of development of our measurement technique.

Reviewer 2

The reviewer writes:

The authors have conceded all of my points, except my misunderstanding of the magnetoresistance. The remaining picture is quite puzzling, however, since all indications are that the system is a good Fermi liquid that is DEEP in the degenerate regime. The statement about uncertainty prescribing the constraint $\tau > 1/k_B T$ does not contain any more or less information than the proposed "Plankian bound," and certainly does not explain why such a time scale can arise in a system that is otherwise well-captured by Fermi liquid theory.

Despite these reservations, my overall inclination is that these results are interesting, and probably worth publishing in Nature Communications, so long as the Editors are aware that these results raise more (potentially very important) questions than they answer.

Response: We fully agree. The results remain puzzling. We completely share the reviewer's interest and look forward to new insights and stimulating discussions in the future.